# Efficacy of 5-Aminolevulinic Acid in Photodynamic Detection and Photodynamic Therapy in Veterinary Medicine

**DOI:** 10.3390/cancers11040495

**Published:** 2019-04-07

**Authors:** Tomohiro Osaki, Inoru Yokoe, Yuji Sunden, Urara Ota, Tomoki Ichikawa, Hideo Imazato, Takuya Ishii, Kiwamu Takahashi, Masahiro Ishizuka, Tohru Tanaka, Liming Li, Masamichi Yamashita, Yusuke Murahata, Takeshi Tsuka, Kazuo Azuma, Norihiko Ito, Tomohiro Imagawa, Yoshiharu Okamoto

**Affiliations:** 1Joint Department of Veterinary Clinical Medicine, Faculty of Agriculture, Tottori University, 4-101 Koyama-Minami, Tottori 680-8553, Japan; inoru.y@gmail.com (I.Y.); sunden@tottori-u.ac.jp (Y.S.); yamashita@muses.tottori-u.ac.jp (M.Y.); ymurahata@muses.tottori-u.ac.jp (Y.M.); tsuka@muses.tottori-u.ac.jp (T.T.); kazu-azuma@muses.tottori-u.ac.jp (K.A.); taromobile@me.com (N.I.); imagawat@muses.tottori-u.ac.jp (T.I.); yokamoto@muses.tottori-u.ac.jp (Y.O.); 2SBI Pharmaceuticals Co., Ltd., Tokyo 106-6020, Japan; uota@sbigroup.co.jp (U.O.); toichika@sbigroup.co.jp (T.I., Tomoki Ichikawa); himazato@sbigroup.co.jp (H.I.); taishii@sbigroup.co.jp (T.I., Takuya Ishii); kiwtakah@sbigroup.co.jp (K.T.); mishizuk@sbigroup.co.jp (M.I.); tortanaka@sbigroup.co.jp (T.T.); 3Department of Bio- and Material Photonics, Chitose Institute of Science and Technology, Chitose 066-0012, Japan; liliming@photon.chitose.ac.jp

**Keywords:** 5-aminolevulinic acid, carcinoma, cat, dog, protoporphyrin IX, sarcoma

## Abstract

5-Aminolevulinic acid (5-ALA), a commonly used photosensitizer in photodynamic detection (PDD) and therapy (PDT), is converted in situ to the established photosensitizer protoporphyrin IX (PpIX) via the heme biosynthetic pathway. To extend 5-ALA-PDT application, we evaluated the PpIX fluorescence induced by exogenous 5-ALA in various veterinary tumors and treated canine and feline tumors. 5-ALA-PDD sensitivity and specificity in the whole sample group for dogs and cats combined were 89.5 and 50%, respectively. Notably, some small tumors disappeared upon 5-ALA-PDT. Although single PDT application was not curative, repeated PDT+/−chemotherapy achieved long-term tumor control. We analyzed the relationship between intracellular PpIX concentration and 5-ALA-PDT in vitro cytotoxicity using various primary tumor cells and determined the correlation between intracellular PpIX concentration and 5-ALA transporter and metabolic enzyme mRNA expression levels. 5-ALA-PDT cytotoxicity in vitro correlated with intracellular PpIX concentration in carcinomas. Ferrochelatase mRNA expression levels strongly negatively correlated with PpIX accumulation, representing the first report of a correlation between mRNA expression related to PpIX accumulation and PpIX concentration in canine tumor cells. Our findings suggested that the results of 5-ALA-PDD might be predictive for 5-ALA-PDT therapeutic effects for carcinomas, with 5-ALA-PDT plus chemotherapy potentially increasing the probability of tumor control in veterinary medicine.

## 1. Introduction

Photodynamics techniques, which utilize photosensitizers that accumulate preferentially in neoplastic and other hyperproliferative tissues, have been recently introduced to clinical practice for both detection and therapeutic purposes. Among these, photodynamic therapy (PDT) constitutes a novel treatment for cancer and certain-malignant conditions. PDT consists of the administration and selective accumulation of a photosensitizer in target tumor tissues, followed by generation of singlet oxygen and other cytotoxic reactive oxygen species that result in oxidative damage to proteins, lipid, and nucleic acids, ultimately leading to cell death by apoptosis and/or necrosis upon irradiation with light of an appropriate wavelength in the presence of tissue oxygen via three main cell death pathways: apoptosis, necrosis, and autophagy [1,2,3,4,5]. Photodynamic detection (PDD) involves utilization the fluorescence of a photosensitizer to localize abnormal tissue and reveals neoplastic lesions that cannot be detected by means of conventional methods [6]. PDT and PDD differ with regard to the wavelength of light applied to the tumor cells and tissues. In PDD, photosensitizers can be efficiently excited by visible blue light (ca. 405 nm) and emit red fluorescence (590–700 nm), whereas in PDT, light administration of approximately 600–800 nm induces phototoxicity and tumor cell death through varying degrees of apoptosis and necrosis [7].

5-Aminolevulinic acid (5-ALA), a commonly used photosensitizer in PDD and PDT, constitutes a precursor of the established photosensitizer protoporphyrin IX (PpIX) that is converted in situ to PpIX via the heme biosynthetic pathway [8,9]. In particular, upon excess 5-ALA administration, the abundantly produced PpIX cannot be converted to heme and thus accumulates within tumor cells [9]. In human medicine, 5-ALA has been used successfully for both PDD and PDT. The use of 5-ALA in PDD has previously been shown to facilitate the visualization of the bladder [6,10,11] along with malignant brain tumors [12,13] and other neoplasms. Furthermore, 5-ALA-PDT has been successfully utilized in the treatment of actinic keratosis [14], Bowen’s disease [15], basal carcinoma, and other diseases [16]. 

In contrast, few reports are available on the clinical evaluation of PDD and PDT using 5-ALA in veterinary medicine, although we recently reported that 5-ALA-PDD might serve as an effective detection tool for canine mammary gland tumors [17]. Several studies have also addressed the efficacy of 5-ALA-PDT to treat feline superficial squamous cell carcinoma, reporting 85% complete remission and 51–64% local recurrence [18,19]. Moreover, the treatment of six dogs with transitional cell carcinoma of the lower urinary tract with 5-ALA-PDT resulted in tumor progression-free intervals from 4 to 34 weeks in five of the animals [20]. In vitro studies have further shown that 5-ALA-PDT destroyed canine transitional cell carcinoma cells [21]. However, little else has been reported regarding the use of 5-ALA-PDT for veterinary tumors.

In this study, in order to extend the range of application of 5-ALA-PDT, we evaluated 5-ALA-PDD in various tumors, then treated dogs and cats with tumors. Moreover, we analyzed the relationship between intracellular PpIX concentration and in vitro cytotoxicity of 5-ALA-PDT using various primary tumor cells and assessed the correlation between intracellular PpIX concentration and mRNA expression levels of 5-ALA transporters and metabolic enzymes.

## 2. Results

### 2.1. Photodynamic Detection (PDD)

The diagnostic accuracy of 5-ALA-PDD was examined in all 144 resected samples of all 124 cases. In 116 malignant tumors, red fluorescence was confirmed in 57 of 64 (Figure 1), 32 of 36, and 14 of 16 subjects with carcinomas, sarcomas, and other tumors, respectively (Table 1). In 28 benign tumors and non-neoplastic lesions, red fluorescence was confirmed in 14 subjects. Figure 2 shows representative images of the positive (Figure 2a), negative (Figure 2b), false-negative (Figure 2c), and false-positive (Figure 2d) cases in 5-ALA-PDD. The sensitivity and specificity were 88.8 and 50%, respectively. 

### 2.2. Photodynamic Therapy (PDT) in Dogs and Cats

Table 2 shows the summary of the subjects treated with 5-ALA-PDT. Of the 14 subjects with tumors (15 tumors), 11 tumors showed red fluorescence induced by 5-ALA. In subjects No. 5, 9, and 11, PDD was not available. Subject No. 14 did not exhibit red fluorescence. Subject No. 9 was orally administered melphalan and prednisolone; subjects No. 10 (Figure 3 and Figure 4) and 11 (Figure 5 and Figure 6) were intra-arterially administered carboplatin and doxorubicin, and numbers (Nos.) 12–14 were orally administered lapatinib. Repeated PDT+/− chemotherapy effectively reduced tumor size and achieved a response rate of 60% (Table 3). 

### 2.3. Photodynamic Therapy (PDT) In Vitro

Figure 7 shows the results of cytotoxicity of 5-ALA-PDT in carcinomas (Figure 7a–i) and sarcomas (Figure 7j–o). Three canine tumor cell lines (IUT, FBC, and YCC) were very sensitive to 5-ALA-PDT. 5-ALA-PDT induced cell death in five canine tumor cell lines (Jack, YDP, HDC, JDM, and ITP) in a 5-ALA concentration-dependent manner. Seven canine tumor cell lines (Gal, LuBi, SNP, SGR, KLC, DML, and HTR) were resistant to 5-ALA-PDT. In carcinomas, the cytotoxicity of 5-ALA-PDT at 1 mM was positively and strongly correlated with the intracellular PpIX concentration (correlation: r = 0.7291, *p* = 0.0129) (Figure 8a), whereas no correlation was observed in sarcomas (r = −0.07622, *p* = 0.4429) (Figure 8b).

### 2.4. Expression of mRNA Associated with PpIX Accumulation

The correlation of PpIX concentration and relative mRNA levels of PEPT1, PEPT2, GAT2, TAUT, PAT1, ABCG2, ABCB6, HMBS, UROS, UROD, CPOX, FECH, and HO-1 in nine carcinoma cell lines (IUT, Jack, Gal, YDP, HDC, LuBi, JDM, FBC, and SNP) and six sarcoma cell lines (SGR, KLC, DML, YCC, ITP, and HTR) are shown in Figure 9 and Figure 10, respectively. In carcinomas, the intracellular PpIX concentration showed a strong negative correlation with the relative *FECH* mRNA levels (r = −0.6368, *p* = 0.0326) (Table 4). In sarcomas, the intracellular PpIX concentration showed a strong negative correlation with the relative *GAT2* mRNA levels (r = −0.7321, *p* = 0.049) (Table 4). The intracellular PpIX concentration showed a strong positive correlation with the relative *UROS* mRNA levels (r = 0.7561, *p* = 0.041) (Table 4).

## 3. Discussion

In the present study we evaluated the use of 5-ALA-PDD in various veterinary tumors as a screen for 5-ALA-PDT and then attempted to extend the range of 5-ALA-PDT application. In the present PDD study, the sample size for each tumor type was small, and size as well as stage of each tumor were different. Therefore, we analyzed the whole sample group. The sensitivity and specificity of 5-ALA-PDT for dogs and cats combined were 89.5 and 50%, respectively. In human medicine, numerous clinical trials have indicated that 5-ALA-PDD improves the detection of bladder cancer and brain tumors. In previous reports regarding bladder cancers, the sensitivity of fluorescence cystoscopy was considered superior to that of white light cystoscopy, with the mean values of sensitivities and specificities for fluorescence cystoscopy being 94.4% (range: 87–97) and 51.3 (range: 35–66.6), respectively [22,23,24,25,26,27,28]. In previous reports regarding brain tumors, 5-ALA-induced PpIX fluorescence improved the results in high grade glioma surgery for gross total resection. PpIX fluorescence induced by 5-ALA was observed in sarcomas as well as carcinomas, with the mean values of sensitivities and specificities for fluorescence cystoscopy being 86.8% (range: 75–92) and 85.2 (range: 71–92), respectively [29,30,31,32,33]. 

The results of PDD in veterinary medicine thus appear similar to those obtained in human medicine. In our study, multiple false positive samples were identified. Accordingly, it was considered that PpIX fluorescence could also be detected in benign tumors and non-neoplastic lesions. In addition, it had previously been reported that false positive fluorescence was related to inflammatory cell infiltration [13], which was consistent with the consideration that some masses with inflammation might have been present in the current study. We also identified some false negative samples. False negative fluorescence might be attributed to structure barriers, such as blood or necrotic tissues [34]. In particular, we did not detect 5-ALA-induced fluorescence in hemangiosarcoma. Previously, it has been suggested that 5-ALA induced fluorescence might be limited in cases of low tumor cell density [35]. In the present study, we did not have a direct comparison for the fluorescence intensity of PpIX of each tumor tissue. However, we previously reported that cell density strongly correlated with PpIX photon counts of mammary gland tumor tissue in dogs [17].

To date, 5-ALA-PDT has been indicated for the treatment of feline superficial squamous cell carcinoma and canine lower urinary tumor [18,19,20]. Based on the results of PDD in the present study, PpIX fluorescence was observed in 88.8% of additional malignant tumors types analyzed. Therefore, we examined the efficacy of 5-ALA-PDT for various tumors in dogs and cats. Tumors exhibiting PpIX fluorescence induced by 5-ALA were responsive to 5-ALA-PDT. Although a single PDT application was not curative, repeated PDT+/− chemotherapy achieved long-term tumor control in these subjects. Conversely, 5-ALA-PDT did not have a therapeutic effect for subject No. 14, in which PpIX fluorescence was not observed. Specifically, PDT combined with intra-arterial chemotherapy was safe, well tolerated, and effective for the treatment of paranasal sinus and lower urinary tract tumors in dogs and may therefore represent a suitable alternative to conventional therapy. 

Finally, to elucidate the mechanism underlying the cytotoxicity of 5-ALA-PDT in vitro, we compared the cytotoxicity and intracellular PpIX concentration using various primary tumor cells. Our in vitro study showed that the cytotoxicity induced by 5-ALA-PDT correlated with intracellular PpIX concentration in carcinomas, which is consistent with a previous report that the cytotoxicity for ovarian clear-cell carcinoma was correlated with the intracellular PpIX [35]. Based on the current findings, it was considered that the therapeutic effects of 5-ALA-PDT for carcinomas were potentially correlated with fluorescence brightness of PpIX induced by 5-ALA in clinical cases. However, we found that LuBi cells, in which PpIX was accumulated at moderate level, were not killed by 5-ALA-PDT. As glutathione peroxidase activity has been reported to contribute to the resistance of tumor cells to PDT [36], it was considered that the 5-ALA-PDT resistance of LuBi cells might be related to such activity. However, the cytotoxicity induced by 5-ALA-PDT did not correlate with intracellular PpIX concentration in sarcomas. Therefore, the cell death mechanism by 5-ALA-PDT in sarcomas remains to be elucidated. In future studies, we will analyze the detailed mechanism underlying the 5-ALA-PDT resistance of these cells. 

In carcinomas, the intracellular PpIX concentration showed a strong negative correlation with the expression of *FECH* mRNA levels but did not show a correlation with the expression of *UROS* mRNA. It has been reported that the expression of FECH protein played an important role in PpIX accumulation in human bladder cancer cells [37]. In addition, the levels of *FECH* mRNA expression in glioblastoma were also found to be lower than those in normal brains [38]. Therefore, the suppression of FECH might contribute to the intracellular accumulation of PpIX in carcinomas. In comparison, in sarcomas, the intracellular PpIX concentration showed a strong positive correlation with the relative *UROS* mRNA levels but did not show a correlation with the expression of *FECH* mRNA. To our knowledge, there have been no reports on the correlation between the relative *UROS* mRNA levels and the intracellular PpIX concentration in human and murine cells. It was considered that the difference in sensitivity to 5-ALA-PDT between carcinomas and sarcomas might be correlated with the differences of the mRNA expression levels of transporters and heme synthesis enzymes of 5-ALA. In future, there is need to be further examined the correlation between the intracellular PpIX concentration and these mRNA levels.

## 4. Materials and Methods

### 4.1. Materials

5-ALA hydrochloride (HCl) was donated by SBI Pharmaceuticals (Tokyo, Japan). 

### 4.2. Photodynamic Detection

#### 4.2.1. Animals

For the PDD study, we enrolled 124 client-owned dogs and cats with masses, which were brought to the Tottori University Veterinary Medical Centre between December 2011 and November 2018. The masses were resected under general anesthesia to provide a definitive diagnosis. A total of 109 dogs (male: 31, castrated male: 14; female: 35, spayed female: 29) were utilized in the age range of 0.9–16 years (mean and median, 10.3 and 11 years, respectively) and weighing between 2.1 and 53 kg (mean and median, 13.2 and 9.6 kg, respectively). In addition, 15 cats (male: 2, castrated male: 3; female: 5, spayed female: 5) were used in the age range of 7–15 years (mean and median, 10.8 and 10 years, respectively) and weighing between 2.7 and 6.6 kg (mean and median, 4.3 and 4.5 kg, respectively). 

#### 4.2.2. PDD Using 5-ALA

The dogs and cats were orally administered 5-ALA HCl at a dose of 40 mg/kg 4 h prior to biopsy or surgery. Generally, butorphanole (0.2 mg/kg) and midazolam (0.15 mg/kg) were administered as pre-anesthetic agents and general anesthesia was induced with intravenous propofol (4 mg/kg), and then maintained with isoflurane and oxygen after tracheal intubation. Furthermore, robenacoxib (Onsior^®^, Elanco Japan, Tokyo, Japan) was administered subcutaneously at a dose of 2 mg/kg as an analgesic agent. After excision of 141 masses with surrounding normal tissues, the masses were further incised through the midline and the non-necrotic areas were fluoresced using an LED light source at 405 nm. The video camera (HDR-CX180, SONY, Tokyo, Japan) was equipped with a long-pass filter designed to block blue light (for observation of fluorescence). For fluorescence spectrometry, LED405-SMA-TI (Thorlabs Japan Inc., Tokyo, Japan), R600-8-UV–VIS-SR (StellarNet, Tampa, FL, USA), and Black-Comet CXR-50 TEC spectrometers (StellarNet) were used to obtain the spectra for each tissue. The maximum fluorescence peak of PpIX was at approximately 635 nm. The PDD protocol was approved by the Ethics Committee at the Faculty of Agriculture, Tottori University (ethical approval number: H28-003).

#### 4.2.3. Histology

To analyze sensitivity and specificity, samples from fluorescent and non-fluorescent tissues were evaluated histologically. The mass tissues were fixed in 4% buffered formalin, embedded in paraffin, sectioned, and stained with hematoxylin and eosin. 

### 4.3. PDT in Dogs and Cats

#### 4.3.1. Animals

For the PDT study, we enrolled 14 client-owned dogs and cats with tumors, which were brought to the Tottori University Veterinary Medical Centre between November 2011 and May 2018. Specifically, 10 dogs (male: 4, castrated male: 1; female: 3, spayed female: 2) were utilized in the age range of 6–15 years (mean and median, 10.3 and 10.5 years, respectively) and weighing between 2.8 and 30 kg (mean and median, 13.9 and 11.5 kg, respectively). In addition, 4 cats (female: 3, spayed female: 1) were used in the age range of 8–13 years (mean and median, 10.7 and 11 years, respectively) and weighing between 3.3 and 4.6 kg (mean and median, 4.1 and 4.4 kg, respectively). Each tumor was diagnosed by histopathological examination of biopsy samples.

#### 4.3.2. PDT Using 5-ALA

A total of 10 dogs and 4 cats with tumors (15 tumors) were orally administered 5-ALA at a dose of 40 mg/kg 4 h prior to irradiation. Simultaneously, the dogs were subcutaneously administered 1 mg/kg maropitant (Zoetis Japan, Tokyo, Japan). The patients were treated under sedation or general anesthesia. For sedation, medetomidine (0.03 mg/kg) and midazolam (0.15 mg/kg) was intramuscularly administered. For general anesthesia, butorphanole (0.2 mg/kg) and midazolam (0.15 mg/kg) were administered as pre-anesthetic agents and general anesthesia was induced with intravenous propofol (4 mg/kg), and then maintained with isoflurane and oxygen after tracheal intubation.

The tumors were irradiated with a Ceralas 630-nm PDT diode laser (Biolitec, Jene, Germany) or 635-nm LED. For superficial lesions, LED and the quartz fiber fitted with a microlens were set toward the surface of the tumor. For interstitial irradiation, an over-the-needle intravenous catheter (14-gauge needle) was inserted at the desired location in the tumor mass. A 1-cm cylindrical diffuser fiber was inserted through an external cylinder of the catheter that had been left in place to provide support to the fiber. The response to PDT was evaluated 1 month after the first PDT according to RECIST criteria. Subsequently, the patients that were responsive to PDT were repeatedly treated by PDT once every week. The PDT protocol was approved by the Ethics Committee at the Faculty of Agriculture, Tottori University (ethical approval number: H28-004). Eight animals (Nos. 1–8) with tumors were treated with PDT only. Six dogs (Nos. 9–14) with tumors were treated with PDT and chemotherapy. 

### 4.4. PDT In Vitro

#### 4.4.1. Primary Cells

The tumor tissues were minced and cultured in 35 mm diameter plastic dishes (Corning Costar, Armonk, NY, USA) in RPMI 1640 medium (Invitrogen, Carlsbad, CA, USA) supplemented with 10% heat-inactivated fetal bovine serum (FBS) (Nichirei Biosciences Inc., Tokyo, Japan) and penicillin-streptomycin-neomycin (PSN) solution (5 mg/mL penicillin, 5 mg/mL streptomycin, and 10 mg/mL neomycin; Invitrogen). The culture dishes were maintained in 5% CO_2_ at 37 °C and observed daily by phase-contrast microscopy. The cells were subcultured by washing with phosphate-buffered saline (PBS) (Nacalai Tesque Inc., Kyoto, Japan) and dispersed in 0.25% trypsin and 1 mmol/L ethylenediaminetetraacetic acid (EDTA) tetrasodium salt solution and phenol red (Invitrogen). Then, the cells were placed into 75 cm^2^ tissue culture flasks (Corning Inc., New York, NY, USA). Cell stocks were stored in a Cellbanker^TM^1 (Nippon Zenyaku Kogyo Co., Ltd., Fukushima, Japan). 

#### 4.4.2. Cell Culture Conditions

A total of 15 canine tumor primary cell lines which were established by author were used in this study (Table 5) [39]. 

All established primary cells were maintained in RPMI 1640 medium supplemented with 10% heat-inactivated FBS and PSN solution. The cells were then incubated in 5% CO_2_ at 37 °C. The primary cells were harvested from near-confluent cultures by brief exposure to a solution containing 0.25% trypsin and 1 mM EDTA tetrasodium salt solution and phenol red. Trypsinization was stopped using RPMI 1640 containing 10% FBS. The cells were centrifuged and resuspended in RPMI 1640. Trypan blue staining was used to assess cell viability. 

#### 4.4.3. Evaluation of the Cytotoxic Effects of Different 5-ALA Doses on Different Primary Cells

We seeded 4–5 × 10^4^ primary cells into each well of 96 well plates (Corning Inc.) and incubated the cells overnight. The different primary cells were then incubated with various concentrations of 5-ALA (0, 0.03, 0.1, 0.3, and 1 mM) for 4 h. After washing with fresh medium, the cells were irradiated with 630 nm light (20 mW/cm^2^, 10 J/cm^2^) emitted by LED lights. Subsequently, the cells were reincubated for 24 h in the dark. Cell viability was examined using a Cell Counting Kit-8 (Dojindo Molecular Technologies, Inc., Kumamoto, Japan) in accordance with the manufacturer’s instructions.

#### 4.4.4. Intracellular PpIX Concentration in Different Primary Cells

We seeded 1–2 × 10^6^ primary cells into a 75 cm^2^ tissue culture flask (Corning Inc.) and incubated the cells overnight. The different primary cells were then incubated with 1 mM 5-ALA for 4 h. The cells were washed twice with PBS and detached from the culture flask by 0.25% trypsin and 1 mM EDTA tetrasodium salt solution and phenol red. The cells were centrifuged, resuspended in PBS to a concentration of 1 × 10^6^/mL, then lysed by sonication for 30 s. For the determination of PpIX, 0.2 mL of homogenate was vigorously agitated for 60 s with 0.02 mL of 50% v/v acetic acid and 0.9 mL of N,N-dimethylformamide-2-propanol solution (100:1 by vol.) and centrifuged at 13,150× *g* for 5 min at 4 °C to collect the supernatant. The supernatants were analyzed with a high performance liquid chromatography system using a Capcell Pak C18 UG120 column (5 μm, 4.6 × 150 mm, Shiseido, Tokyo, Japan), mobile phase of acetonitrile-10 mM tetrabutylammonium hydroxide solution (pH 7.5) (70:30 by vol., flow rate, 1.0 mL/min; elution temperature, 40 °C), and fluorescence detector (Ex. 400 nm, Em. 630 nm).

#### 4.4.5. Detection of mRNA Levels by Quantitative Real-Time Reverse Transcription-Polymerase Chain Reaction (qRT-PCR)

Total RNA extraction was performed using an RNeasy Mini Kit (Qiagen, Hilden, Germany) according to the manufacturer’s instructions. The RNA concentration was determined using a Nanodrop One microvolume spectrophotometer (Thermo Fisher Scientific, Waltham, MA, USA). Transcription was performed using a High-capacity RNA-to-cDNA kit (Applied Biosystems, Waltham, MA, USA) according to the manufacturer’s instructions. To obtain cDNA, 1 μg of RNA was used, and the resulting cDNA was stored at −20 °C. The RNA levels of peptide transporter 1 (PEPT1), peptide transporter 2 (PEPT2), GABA transporter 2 (GAT2), taurine transporter (TAUT), H+/amino acid transporter 1 (PAT1), ATP-binding cassette transporter G2 (ABCG2), ATP-binding cassette transporter B6 (ABCB6), hydroxymethylbilane synthase (HMBS), uroporphyrinogen III synthase (UROS), uroporphyrinogen III decarboxylase (UROD), coproporphyrinogen oxidase (CPOX), ferrochelatase (FECH), heme oxigenase 1 (HO-1), and β-actin were determined by qRT-PCR on the FI Step One Plus^TM^ system using a SYBR Select Master Mix (Applied Biosystems®, Waltham, MA, USA). β-Actin was used as the internal control, and 20 ng of cDNA was used for qRT-PCR. The cycling conditions consisted of a 2-min hot start at 50 and 95 °C, followed by 40 cycles of denaturation at 95 °C for 15 s, annealing at 60 °C for 60 s, extension at 95 °C for 15 s, and then a final inactivation at 95 °C for 15 s. Relative quantification was performed using the ΔΔCT method. The specific primers for each gene are shown in Table 6. The selection of mRNA was performed according to the methods as described [35]. All experiments were performed in triplicate. 

### 4.5. Statistical Analysis

The statistical analyses were performed using GraphPad Prism version 6 (GraphPad Software, La Jolla, CA, USA). A *p*-value < 0.05 was considered statistically significant. The statistical correlations were calculated using the Pearson correlation. 

## 5. Conclusions

The results of the present studies expand the utilization of photodynamics methodologies within veterinary medicine, which in turn may facilitate translational studies using animal models addressed at diagnosing and treating human disorders. Furthermore, the present study is the first to demonstrate a correlation between the expression of mRNA involved in PpIX accumulation and PpIX concentration in canine tumor cells. In this study, we evaluated the expression level of each mRNA using qRT-PCR. 5-ALA-PDT cytotoxicity in vitro correlated with intracellular PpIX concentration in carcinomas. *FECH* mRNA expression levels in carcinomas strongly negatively correlated with PpIX accumulation. Conversely, the cytotoxicity induced by 5-ALA-PDT did not correlate with intracellular PpIX concentration in sarcomas. To validate these results, the expression of each protein would also need to be evaluated by western blotting. Future experiments are also required that utilize an inhibitor of ABC family transporters and/or knockdown experiments using siRNA against heme synthesis pathway components.

## Figures and Tables

**Figure 1 cancers-11-00495-f001:**
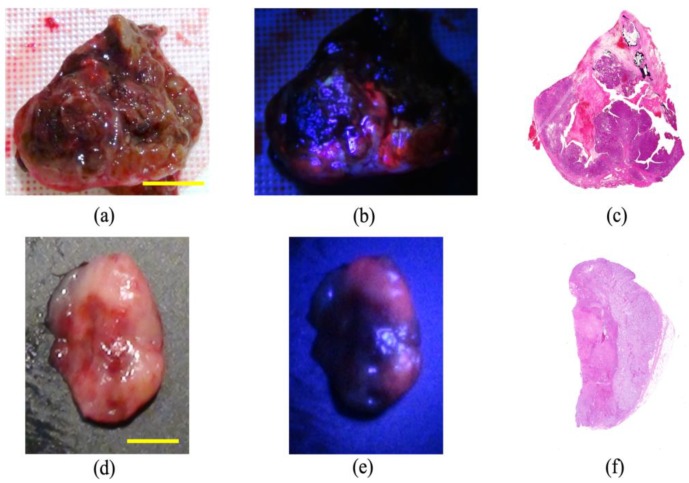
Representative images of photodynamic detection-positive cases. (**a**–**c**) Canine thyroid carcinoma. The red fluorescence at 635 nm is evident in the cancerous lesions but not in the necrotic tumor lesions. Scale bar: 2 cm. (**d**–**f**) Inguinal lymph node metastasis in a mammary gland tumor. The red fluorescence at 635 nm is evident in the metastatic lesions but not in the non-metastatic regions. Scale bar: 3 mm. (**a**,**d**) White light images; (**b**,**e**) fluorescence light images; (**c**,**f**) hematoxylin and eosin-stained images.

**Figure 2 cancers-11-00495-f002:**
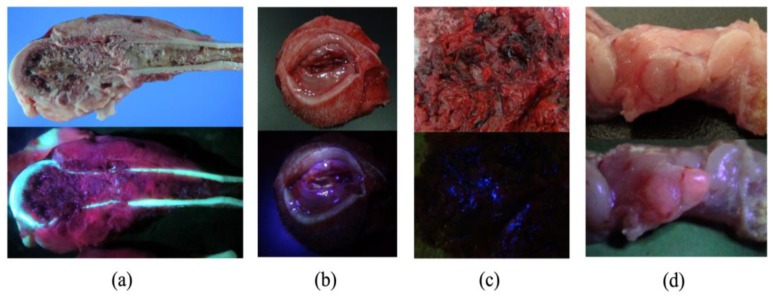
Representative images of photodynamic detection cases. The upper images comprise the white light images and the lower images are the fluorescence images. (**a**) Positive case of osteosarcoma of the distal femur. The red fluorescence at 635 nm is evident in the cancerous lesions but not in the necrotic tumor lesions. (**b**) Negative case of a subcutaneous granulomatous inflammation. Red fluorescence was not observed. (**c**) False-negative case of splenic hemangiosarcoma with a large quantity of blood. Although tumor cells were observed by histological examination, there was no peak for photosensitizer protoporphyrin IX (PpIX) in the fluorescence spectrum. (**d**) False-positive case of lymphatic follicles. The red fluorescence and follicles occur at the same location, which might be related to the infiltration of inflammatory cells.

**Figure 3 cancers-11-00495-f003:**
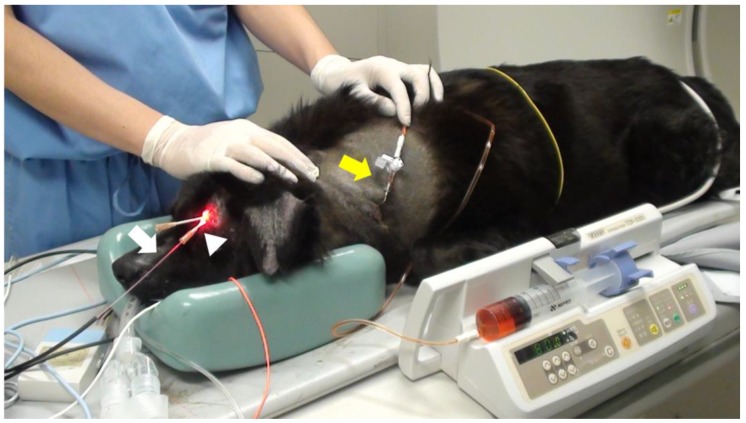
Interstitial photodynamic therapy and intra-arterial chemotherapy injection for case No. 10. A 12 year old, mixed-breed dog presented with swelling of the left frontal sinus. The dog was orally administered 40 mg/kg of 5-ALA. After 4 h, the tumor was interstitially irradiated with 630 nm laser light emitted (150 mW, 270 J) by a diode laser under general anesthesia. The dog also received weekly doses of 100 mg/m^2^ carboplatin and 10 mg/m^2^ doxorubicin. An over-the-needle intravenous catheter was inserted through the skin directly into the tumor under computed tomography imaging guidance. The cylindrical fiber (white arrow) could then be passed through an external cylinder of the catheter (white arrowhead). The fiber was placed inside a 14 gauge needle and inserted into the tissue. Doxorubicin hydrochloride was administered with a Huber needle through the skin into the port (yellow arrow).

**Figure 4 cancers-11-00495-f004:**
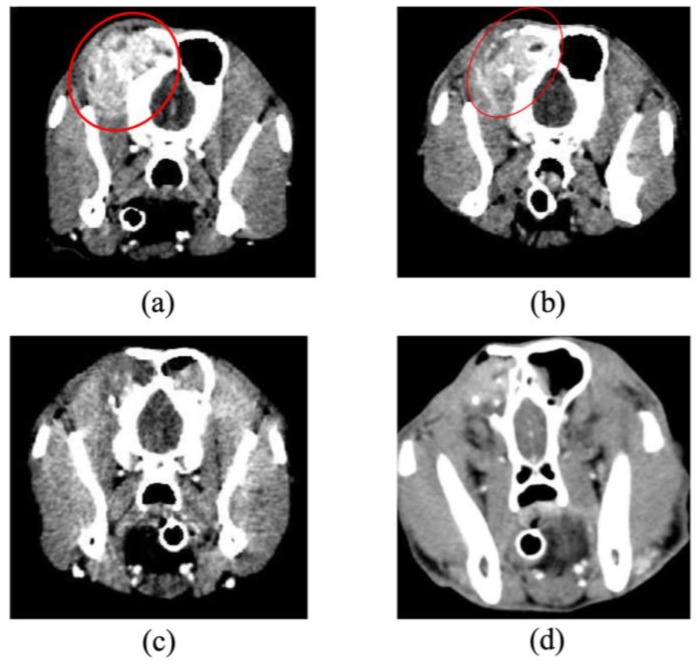
Angiographic computed tomography images of case No. 10. (**a**) The dog was diagnosed with adenocarcinoma of the left paranasal sinus on Day 1. Opacities consistent with soft tissue could be observed in the images of the left nasal cavity and frontal sinus, and the forehead bone was significantly absorbed (red circle). (**b**) The tumor slightly decreased in size on Day 14 (red circle). (**c**) Complete remission was achieved on Day 56. (**d**) Imaging and histological evaluation showed no progression of disease during treatment on Day 304. The dog survived for 718 days after initial presentation.

**Figure 5 cancers-11-00495-f005:**
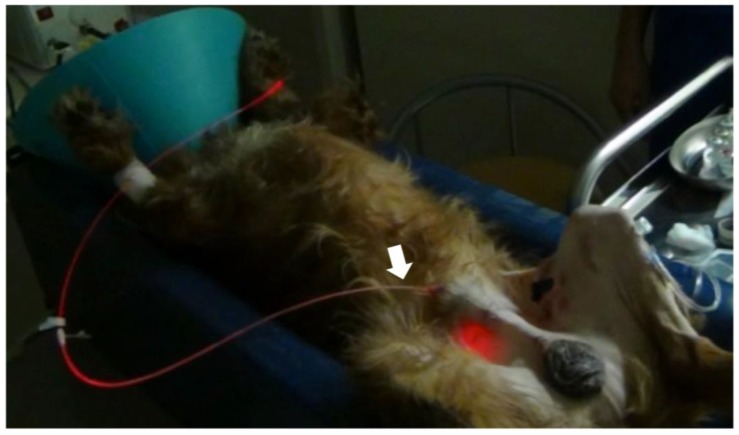
Photodynamic therapy and intra-arterial chemotherapy injection for case No. 11. An 11 year old, mixed-breed dog presented with hematuria and polyuria. The dog was diagnosed with transitional cell carcinoma of the lower urinary tract and treated with photodynamic therapy as in case No. 10 along with intra-arterial chemotherapy. An optical fiber placed in an 8Fr versatile catheter was inserted into the urethra (white arrow). The fiber tip was placed at the tumor site under ultrasound and the tumor was irradiated (150 mW, 270 J) under sedation.

**Figure 6 cancers-11-00495-f006:**
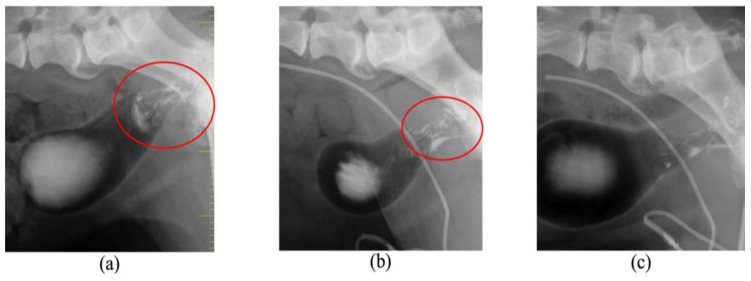
Double contrast images of case No. 11. (**a**) The tumor size of approximately 2 cm in diameter at the trigone of the bladder could be observed on Day 1 (red circle). The tumor slightly decreased in size on Day 56 (red circle) (**b**) and was markedly reduced on day 196 (**c**). The dog survived for 393 days after initial presentation.

**Figure 7 cancers-11-00495-f007:**
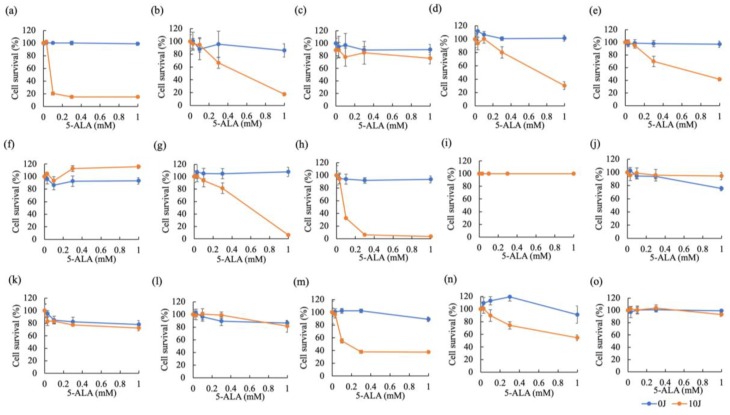
Cytotoxicity of 5-ALA-photodynamic therapy (PDT). A total of 15 primary tumor cell lines (Carcinomas: (**a**) IUT, (**b**) Jack, (**c**) Gal, (**d**) YDP, (**e**) HDC, (**f**) LuBi, (**g**) JDM, (**h**) FBC, and (**i**) SNP; Sarcomas: (**j**) SGR, (**k**) KLC, (**l**) DML, (**m**) YCC, (**n**) ITP, and (**o**) HTR) were incubated with different concentrations of ALA (0, 0.03, 0.1, 0.3, 1 mM) and irradiated for 0 or 500 s. Cell viability was examined using Cell Counting Kit 8 (Dojindo Molecular Technologies, Inc., Kumamoto, Japan). Data are expressed as the means ± S.D. (*n* = 6).

**Figure 8 cancers-11-00495-f008:**
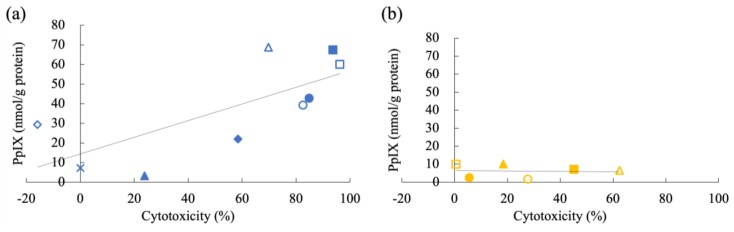
Relationship between the effects of photodynamic therapy (PDT) and the intracellular PpIX concentration in carcinomas (**a**) and sarcomas (**b**). Carcinomas: IUT (blue closed circle), Jack (blue open circle), Gal (blue closed triangle), YDP (blue open triangle), HDC (blue closed diamond), LuBi (blue open diamond), JDM (blue closed square), FBC (blue open square), and SNP (blue cross); Sarcomas: SGR (orange closed circle), KLC (orange open circle), DML (orange closed triangle), YCC (orange open triangle), ITP (orange closed square), and HTR (orange open square). Correlation of 5-ALA-PDT cytotoxicity at 1 mM with intracellular PpIX concentration was determined using Pearson correlation analysis.

**Figure 9 cancers-11-00495-f009:**
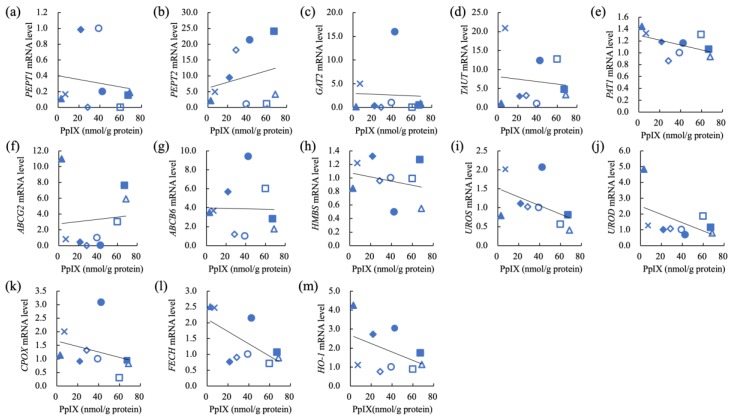
Correlation of intracellular PpIX concentration and relative mRNA levels in carcinomas. (**a**) Peptide transporter 1 (PEPT1), (**b**) peptide transporter 2 (PEPT2), (**c**) GABA transporter 2 (GAT2), (**d**) taurine transporter (TAUT), (**e**) H+/amino acid transporter 1 (PAT1), (**f**) ATP-binding cassette transporter G2 (ABCG2), (**g**) ATP-binding cassette transporter B6 (ABCB6, (**h**) hydroxymethylbilane synthase (HMBS), (**i**) uroporphyrinogen III synthase (UROS), (**j**) uroporphyrinogen III decarboxylase (UROD), (**k**) coproporphyrinogen oxidase (CPOX), (**l**) ferrochelatase (FECH), (**m**) heme oxigenase 1 (HO-1). Carcinomas: IUT (blue closed circle), Jack (blue open circle), Gal (blue closed triangle), YDP (blue open triangle), HDC (blue closed diamond), LuBi (blue open diamond), JDM (blue closed square), FBC (blue open square), and SNP (blue cross); Sarcomas: SGR (orange closed circle), KLC (orange open circle), DML (orange closed triangle), YCC (orange open triangle), ITP (orange closed square), and HTR (orange open square). Pearson’s correlation analysis (*n* = 9).

**Figure 10 cancers-11-00495-f010:**
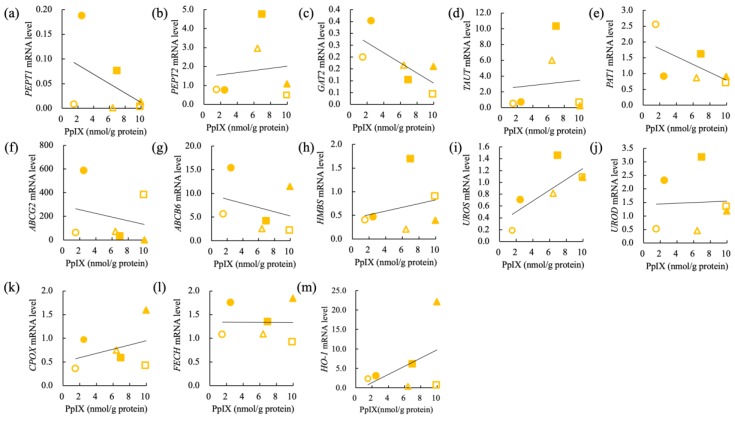
Correlation of intracellular PpIX concentration and relative mRNA levels in sarcomas. (**a**) Peptide transporter 1 (PEPT1), (**b**) peptide transporter 2 (PEPT2), (**c**) GABA transporter 2 (GAT2), (**d**) taurine transporter (TAUT), (**e**) H+/amino acid transporter 1 (PAT1), (**f**) ATP-binding cassette transporter G2 (ABCG2), (**g**) ATP-binding cassette transporter B6 (ABCB6), (**h**) hydroxymethylbilane synthase (HMBS), (**i**) uroporphyrinogen III synthase (UROS), (**j**) uroporphyrinogen III decarboxylase (UROD), (**k**) coproporphyrinogen oxidase (CPOX), (**l**) ferrochelatase (FECH), (**m**) heme oxigenase 1 (HO-1). Carcinomas: IUT (blue closed circle), Jack (blue open circle), Gal (blue closed triangle), YDP (blue open triangle), HDC (blue closed diamond), LuBi (blue open diamond), JDM (blue closed square), FBC (blue open square), and SNP (blue cross); Sarcomas: SGR (orange closed circle), KLC (orange open circle), DML (orange closed triangle), YCC (orange open triangle), ITP (orange closed square), and HTR (orange open square). Pearson’s correlation analysis (*n* = 6).

**Table 1 cancers-11-00495-t001:** Summary of the results of photodynamic detection using 5-aminolevulinic acid.

Malignant Tumors (103/116) ^1^			Benign Tumors (8/14)	Non-Neoplastic Lesions (6/14)
Carcinomas (57/64)	Sarcomas (32/36)	Other Tumors (14/16)		
Mammary gland tumor (29/32)	Osteosarcoma (7/8)	Mast cell tumor (11/13)	Skin appendage tumor (2/3)	Lymphadenopathy (1/4)
Transitional cell carcinoma (5/5)	Leiomyosarcoma (5/5)	Lymphoma (3/3)	Lipoma (0/2)	Nodular hyperplasia (2/2)
Adenocarcinoma (5/5)	Melanoma (5/5)		Sebaceous gland tumor (1/1)	Adenomatous hyperplasia (1/1)
Sertoli cell tumor (2/4)	Liposarcoma (5/5)		Histiocytoma (1/1)	Cystitis (1/2)
Thyroid carcinoma (3/3)	Hemangiosarcoma (2/5)		Plasmacytoma (1/1)	Panniculitis (1/2)
Schwanoma (2/3)	Hemangiopericytoma (3/3)		Trichoepithelioma (1/1)	Granuloma (0/1)
Hepatocellular carcinoma (3/3)	Sarcoma (1/1)		Transitional cell papilloma (1/1)	Rhinitis (0/1)
Squamous cell carcinoma (3/3)	Chondrosarcoma (1/1)		Granular cell tumor (1/1)	Testitis (0/1)
Perianal gland tumor (1/1)	Fibrosarcoma (1/1)		Fibroma (0/1)	
Salivary gland carcinoma (1/1)	Gastrointestinal stromal tumor (1/1)		Leiomyoma (0/1)	
Malignant trichoepithelioma (1/1)	Histiocytic sarcoma (1/1)		Myelolipoma (0/1)	
Sebaceous gland carcinoma (1/1)				
Glioblastoma (1/1)				
Astrocytoma (0/1)				

^1^ Numbers in parentheses: Number of positive subjects/number of positive and negative subjects.

**Table 2 cancers-11-00495-t002:** Summary of the subjects treated with photodynamic therapy using 5-aminolevulinic acid.

No.	Species	Breed	Age (y)	Sex ^1^	Weight (kg)	Tumor Type	Tumor Site	Simultaneous Therapy	Fluorescence
1	Dog	Minatare Schnauzer	6	M	8	Histiocytoma	Nasal planum		+
2	Dog	Dogo Argentino	9	F	30	Mast cell tumor/Mast cell tumor	Skin		+/+
3	Dog	Chihuahua	12	SF	2.8	Transitional cell carcinoma	Nasal cavity		+
4	Cat	American shorthair	11	F	3.3	Sebaceous gland carcinoma	Skin		+
5	Cat	Mix	11	SF	4.3	Fibrosarcoma	Subcutis		N.A. ^2^
6	Cat	Mix	8	F	4.6	Mammary gland tumor	Mammary gland		+
7	Cat	Mix	13	F	4.5	Mammary gland tumor	Lymph node		+
8	Dog	Mix	15	M	6.8	Hemangiopericytoma	Skin		+
9	Dog	Golden Retriever	6	CM	30	Myeloma	Calcaneus	Melphalan	N.A.
10	Dog	Mix	12	F	21.3	Adenocarcinoma	Paranasal sinus	Doxorubicin, Carboplatin	+
11	Dog	Mix	11	M	17.4	Transitional cell carcinoma	Lower urinary tract	Doxorubicin, Carboplatin	N.A.
12	Dog	Minatare Dachshund	13	SF	4.2	Adenocarcinoma	Nasal cavity	Lapatinib	+
13	Dog	Mix	10	M	15	Chondrosarcoma	Nasal cavity	Lapatinib	+
14	Dog	Minatare Dachshund	9	F	3.6	Inflammatory mammary gland tumor	Mammary gland	Lapatinib	-

^1^ M, male; CM, castrated male; F, female; SF, spayed female; ^2^ N.A.: not available.

**Table 3 cancers-11-00495-t003:** Response of tumors to photodynamic therapy using 5-aminolevulinic acid.

No.	Light Equipment	Light Intensity (mW/cm^2^) ^1^ (mW/cm) ^2^		Total Light Fluence (J/cm^2^) ^3^ (J/cm) ^4^		Number of Treatments	Response ^5^	Survival Time/Follow-up Period	Outcome
1	LED	200		240		5	CR	82	Alive
2	LED	200		60		1/1	CR/SD	47/39	-/Surgical extraction
3	LED	200		300		3	PR	75	PDT with another PS ^6^
4	LED	200		120		4	PR	70	PDT with another PS
5	Diode laser	150		270		5	SD	74	Surgical extraction
6	Diode laser	50		20		3	PR	14	Died (metastasis)
7	Diode laser		200		200	4	SD	114	Euthanized
8	Diode laser		300		200	11	PR	58	Surgical extraction
9	Diode laser	200		270		16	PD	334	Died (progress of disease)
10	Diode laser	150		270		15	CR	718	Died (progression of brain tumor)
11	Diode laser	150		270		15	PR	393	Died (old age)
12	Diode laser		300		700	19	PR	355	Treatment interruption
13	Diode laser		320		800	16	PD	232	Died (progress of disease)
14	Diode laser		250		1,000	5	PD	10	Treatment interruption

^1^ Light intensity when tumors were superficially irradiated; ^2^ Light intensity when tumors were interstitially irradiated; ^3^ Light fluence when tumors were superficially irradiated; ^4^ Light fluence when tumors were interstitially irradiated; ^5^ CR: complete response, PR: partial response, SD: stable disease, PD: progressive disease; ^6^ PDT with another PS: Photodynamic therapy with another photosensitizer.

**Table 4 cancers-11-00495-t004:** Correlation of PpIX concentration and relative mRNA levels in carcinomas and sarcomas.

**PpIX vs.**		**PEPT1**	**PEPT2**	**GAT2**	**TAUT**	**PAT1**	**ABCG2**	**ABCB6**
Carcinomas	r	−0.1493	0.2446	−0.03814	−0.1145	−0.5074	0.08762	−0.02171
	R square	0.02230	0.05985	0.001454	0.01312	0.2575	0.007678	0.0004713
	P values	0.3507	0.2629	0.4612	0.3846	0.0816	0.4113	0.4779
Sarcomas	r	−0.4824	0.1172	−0.7321	0.09674	−0.6289	−0.2354	−0.2986
	R square	0.2327	0.01374	0.5360	0.009359	0.3955	0.05540	0.08914
	P values	0.1663	0.4125	0.0490	0.4277	0.0905	0.3267	0.2827
**PpIX vs.**		**HMBS**	**UROS**	**UROD**	**CPOX**	**FECH**	**HO-1**	
Carcinomas	r	−0.2590	−0.4702	−0.4961	−0.3044	−0.6368	−0.4461	
	R square	0.06709	0.2211	0.2461	0.09268	0.4055	0.1990	
	P values	0.2505	0.1007	0.0872	0.2129	0.0326	0.1144	
Sarcomas	r	0.2535	0.7561	0.04538	0.3548	−0.001319	0.4609	
	R square	0.06428	0.5717	0.002059	0.1259	1.739e-006	0.2124	
	P values	0.3139	0.0410	0.4660	0.2450	0.4990	0.1788	

Peptide transporter 1 (PEPT1), peptide transporter 2 (PEPT2), GABA transporter 2 (GAT2), taurine transporter (TAUT), H+/amino acid transporter 1 (PAT1), ATP-binding cassette transporter G2 (ABCG2), ATP-binding cassette transporter B6 (ABCB6), hydroxymethylbilane synthase (HMBS), uroporphyrinogen III synthase (UROS), uroporphyrinogen III decarboxylase (UROD), coproporphyrinogen oxidase (CPOX), ferrochelatase (FECH), and heme oxigenase 1 (HO-1).

**Table 5 cancers-11-00495-t005:** Primary tumor cell lines used in this study.

No.	Primary Tumor Cell Name	Tumor Type (Origin)	Passage
**Carcinomas**			
1	INT	Basal cell carcinoma (peritoneal dissemination)	8
2	Jack	Prostate cancer (Prostate)	14
3	Gal	Squamous cell carcinoma (Nasal cavity)	16
4	YDP	Transitional cell carcinoma (Bladder)	9
5	HDC	Adenocarcinoma (Lung)	8
6	LuBi	Adenocarcinoma (Lung)	20
7	JDM	Adenocarcinoma (Lung)	25
8	FBC	Adenocarcinoma (Nasal cavity)	20
9	SNP	Mammary gland tumor (Malignant pleural effusion)	91
**Sarcomas**			
10	SGR	Osteosarcoma (Tibia)	17
11	KLC	Chondrosarcoma (Nasal cavity)	9
12	DML	Liposarcoma (Subcutis)	60
13	YCC	Melanoma (Oral cavity)	10
14	ITP	Melanoma (Oral cavity)	6
15	HTR	Melanoma (Oral cavity)	12

**Table 6 cancers-11-00495-t006:** Specific primers for each gene.

Primer	Sequence	Primer	Sequence
PEPT1 (NM_001003036)	Fw: TTCTCCAATCATCCAACCTTGA Rv: TGCTCTGCGTTCCGATTACA	HMBS (XM_014113375.1)	Fw: CATGTATGCTGTGGGTCAGG Rv: CAGGTACAGTTGCCCATCCT
PEPT2 (XM_545128)	Fw: CAAACGGTACACACAGTCCTATCAT Rv: TGATACCTCCTGTTCCCAAAGC	UROS (XM_005637839.2)	Fw: TGGCCAAGATCCATACATCA Rv: ACAGCTCCACTGCTTCCACT
GAT2 (XM_003639941)	Fw: TCATTCTTGGCGTGTCTGTCA Rv: ACATACCGCCCTCTGTAAGCA	UROD (XM_005629077.2)	Fw: GCAGTGGCCTCTGAACTAGG Rv: ATTCGAAGCAGCTGGTGACT
TAUT (NM_001003311)	Fw: TGGCGCAGGCATCAAGTT Rv: GATGGCATAGGAGAAGAATATTTGG	CPOX (XM_545070)	Fw: CGCGCGATGGGAGTACAT Rv: TCGGATGGCGCAGAACTT
PAT1 (XM_536073)	Fw: CAGCGTGTCGCCATTGC Rv: GCGCTGATGTTGCCTCATC	FECH (XM_847843)	Fw: TGGACCGAGACCTCATGACA Rv: GGCGTTTGGCGATGATTG
ABCG2 (NM_001048021)	Fw: TCTGGATGAGCCCACAACTG Rv: TTCAGGAGCAAAAGGACAGCAT	HO-1 (NM_001194969)	Fw: GGCAGAGGGTCATCGAAGAG Rv: CTCCTCAAACAGCTGAATGTTCA
ABCB6 (XM_536073)	Fw: CAGCGTGTCGCCATTGC Rv: GCGCTGATGTTGCCTCATC	β-actin (NM_001195845)	Fw: GGCACCCAGCACAATGAAG Rv: GAGCCCCCAATCCACACA

Peptide transporter 1 (PEPT1), peptide transporter 2 (PEPT2), GABA transporter 2 (GAT2), taurine transporter (TAUT), H+/amino acid transporter 1 (PAT1), ATP-binding cassette transporter G2 (ABCG2), ATP-binding cassette transporter B6 (ABCB6), hydroxymethylbilane synthase (HMBS), uroporphyrinogen III synthase (UROS), uroporphyrinogen III decarboxylase (UROD), coproporphyrinogen oxidase (CPOX), ferrochelatase (FECH), and heme oxigenase 1 (HO-1).

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
