# Peer review of "Efficacy of 5-Aminolevulinic Acid in Photodynamic Detection and Photodynamic Therapy in Veterinary Medicine"

_cancers, 2019, doi:10.3390/cancers11040495_

Round 1
Reviewer 1 Report
Interesting article showing the use of 5-ALA to detect and potentially treat cancers in dogs and cats.
Abstract
The abstract doesn't follow the order of the manuscript. I recommend reordering the sentences to focus on the in vivo findings followed by the in vitro findings, as you do in the paper.
Lines 28-29: Reword to make it more clear that data from dogs and cats were combined together and then sensitivity and sensitivity were calculated using histopathology as a gold standard.
Lines 30-31: The Tables indicate progressive disease was observed following PDT +/- chemo treatments in at least 3/14 patients. Please reword the sentence so that is it more accurate.
Lines 35-37: The concluding sentence of the abstract overstates your findings. Please reword to be more specific about which tumor types you did or did not see a correlation in.
Results
Lines 85-86: Suggested rewording = "... of positive (Fig. 2a), negative (Fig. 2b), false-negative (Fig. 2c), and false-positive (Fig. 2d) cases using 5-ADD-PDD."
Lines 179-180: Indicate why you chose to evaluate the mRNAs that you detected.
Tables don't typically have numbered footnotes. I suggest you ask for help editing the Tables to conform to the style preferred by the journal.
Table 2 & 3: Suggested column title change = "Number of Treatments".
It is difficult to flip back and forth between Table 2 &3 to see tumor type, species, treatment, and response to treatment together. This is important to do because different tumors have drastically different expected outcomes even without treatment. For example, histiocytomas regress spontaneously without treatment and grade I mast cell tumors that are completely excised are not expected to recur. Combining these Tables would be beneficial. Please elaborate about the Outcome of a brain tumor in patient #10.
Figures. Please make it clear in the figure legends that each point on the graphs in Figs. 8-10 represents a different cell line.
Discussion
Line 235: Density is used twice.
Lines 263-275: Did any of your mRNA findings contradict your expectations? Is there a way to expand this section of the discussion to explain why most mRNA levels did not correlate with PpIX concentrations?
M&M
Lines 352-359: The information in this paragraph is well represented in the Table already and could be removed.
Conclusions: Indicate which type of cancer correlated and which did not.
Author Response
Responses to Reviewer comments
Reviewer 1
Interesting article showing the use of 5-ALA to detect and potentially treat cancers in dogs and cats.
Abstract
The abstract doesn't follow the order of the manuscript. I recommend reordering the sentences to focus on the in vivo findings followed by the in vitro findings, as you do in the paper.
Response: We appreciate this thoughtful suggestion and have reordered the Abstract as recommended.
Lines 28-29: Reword to make it more clear that data from dogs and cats were combined together and then sensitivity and sensitivity were calculated using histopathology as a gold standard.
Response:We thank the Reviewer for this helpful recommendation. We have added “in the whole sample group” to this sentence to clarify this issue.
Lines 30-31: The Tables indicate progressive disease was observed following PDT +/- chemo treatments in at least 3/14 patients. Please reword the sentence so that is it more accurate.
Response: We thank the Reviewer for the careful assessment of our data. We have reworded the sentence and throughout the manuscript as follows.
“complete or partial remission”→“long-term tumor control”
Lines 35-37: The concluding sentence of the abstract overstates your findings. Please reword to be more specific about which tumor types you did or did not see a correlation in.
Response: We appreciate this advice and have added “for carcinomas” to the Abstract to clarify this point.
Results
Lines 85-86: Suggested rewording = "... of positive (Fig. 2a), negative (Fig. 2b), false-negative (Fig. 2c), and false-positive (Fig. 2d) cases using 5-ADD-PDD."
Response:We thank the Reviewer for this specific suggestion and have substituted “false” for “pseudo” as indicated.
Lines 179-180: Indicate why you chose to evaluate the mRNAs that you detected.
Response: We agree with the Reviewer that this is an important question. This was performed with reference to a previous study (reference number 35). We have added a sentence in the revised Materials & Methods section 4.4.5. as follows.
“The selection of mRNA was performed according to the methods as described" [35].”
Tables don't typically have numbered footnotes. I suggest you ask for help editing the Tables to conform to the style preferred by the journal.
Response: We thank the Reviewer for this suggestion. We note, however, that the example Table included in the template provided by Cancers uses numbered footnotes; we have therefore incorporated this usage.
Table 2 & 3: Suggested column title change = "Number of Treatments".
Response:We appreciate this suggestion and have changed the column title as indicated.
It is difficult to flip back and forth between Table 2 &3 to see tumor type, species, treatment, and response to treatment together. This is important to do because different tumors have drastically different expected outcomes even without treatment. For example, histiocytomas regress spontaneously without treatment and grade I mast cell tumors that are completely excised are not expected to recur. Combining these Tables would be beneficial.
Response:We are grateful for this thoughtful suggestion. However, although we also would like to combine these Tables, the combined Table is too large to be readily accommodated in the manuscript template.
Please elaborate about the Outcome of a brain tumor in patient #10.
Response:We thank the Reviewer for this suggestion. We have explained the outcome of the brain tumor in patient #10 as follows.
“Died (progression of brain tumor)”
Figures. Please make it clear in the figure legends that each point on the graphs in Figs. 8-10 represents a different cell line.
Response: We appreciate this thoughtful suggestion. We have made it clear in the figure legends that each point on the graphs in Figs. 8–10 represents a different cell line.
Discussion
Line 235: Density is used twice.
Response:We thank the Reviewer for bringing this to our attention and have removed the duplicated term.
Lines 263-275: Did any of your mRNA findings contradict your expectations? Is there a way to expand this section of the discussion to explain why most mRNA levels did not correlate with PpIX concentrations?
Response: We thank the Reviewer for this suggestion. As describe in discussion, the intracellular PpIX concentration did not show a correlation with the expression of FECHmRNAin sarcomas.We have added “In future, there is need to be further examined the correlation betweenthe intracellular PpIX concentration and these mRNA levels.” to this sentence to clarify this issue.
M&M
Lines 352-359: The information in this paragraph is well represented in the Table already and could be removed.
Response: We thank the Reviewer for this suggestion. We have removed the indicated sentences from the revised manuscript.
Conclusions: Indicate which type of cancer correlated and which did not.
Response: We appreciate this suggestion. We have re-written the sentence as follows.
“5-ALA-PDT cytotoxicity in vitro correlated with intracellular PpIX concentration in carcinomas. FECHmRNA expression levels in carcinomas strongly negatively correlated with PpIX accumulation. Conversely, the cytotoxicity induced by 5-ALA-PDT did not correlate with intracellular PpIX concentration in sarcomas.”
Reviewer 2 Report
The paper entitled „Efficacy of 5-Aminolevulinic Acid in Photodynamic Detection and Photodynamic Therapy in Veterinary Medicine” by Tomohiro Osaki, Inoru Yokoe, Yuji Sunden, Urara Ota , Tomoki Ichikawa, Hideo Imazato, Takuya Ishii, Kiwamu Takahashi, Masahiro Ishizuka, Tohru Tanaka, Liming Li, Masamichi Yamashita, Yusuke Murahata, Takeshi Tsuka, Kazuo Azuma, Norihiko Ito, Tomohiro Imagawa and Yoshiharu Okamoto reports a correlation between the expression of mRNA involved in PpIX accumulation and PpIX concentration in canine tumor cells. The paper does not opens-up a new area. However, obtained data significantly improve our knowledge about ALA-PDT. In this form article can be accepted for publication in Cancers. Authors should consider re-writing sentence (line 52): “In PDD, the fluorescence emission wavelength is generally set at 405 nm….”. This is confusing It can be concluded that at 405 nm fluorescence signal is measured, whereas it is excitation wavelength.
Author Response
Responses to Reviewer comments
Reviewer 2
The paper entitled „Efficacy of 5-Aminolevulinic Acid in Photodynamic Detection and Photodynamic Therapy in Veterinary Medicine” by Tomohiro Osaki, Inoru Yokoe, Yuji Sunden, Urara Ota , Tomoki Ichikawa, Hideo Imazato, Takuya Ishii, Kiwamu Takahashi, Masahiro Ishizuka, Tohru Tanaka, Liming Li, Masamichi Yamashita, Yusuke Murahata, Takeshi Tsuka, Kazuo Azuma, Norihiko Ito, Tomohiro Imagawa and Yoshiharu Okamoto reports a correlation between the expression of mRNA involved in PpIX accumulation and PpIX concentration in canine tumor cells. The paper does not opens-up a new area. However, obtained data significantly improve our knowledge about ALA-PDT. In this form article can be accepted for publication in Cancers.
Authors should consider re-writing sentence (line 52): “In PDD, the fluorescence emission wavelength is generally set at 405 nm….”. This is confusing It can be concluded that at 405 nm fluorescence signal is measured, whereas it is excitation wavelength.
Response: We thank the Reviewer for the supportive and critical comments and for the helpful suggestion. We have re-written the sentence in the revised manuscript as follows.
“In PDD, photosensitizers can be efficiently excited by visible blue light (ca. 405 nm) and emit red fluorescence (590–700 nm), whereas in PDT, light administration of approximately 600–800 nm induces phototoxicity and tumor cell death through varying degrees of apoptosis and necrosis [4,6,7].”